# Biological Scaffolds for Congenital Heart Disease

**DOI:** 10.3390/bioengineering10010057

**Published:** 2023-01-02

**Authors:** Amy G. Harris, Tasneem Salih, Mohamed T. Ghorbel, Massimo Caputo, Giovanni Biglino, Michele Carrabba

**Affiliations:** 1Bristol Heart Institute, Bristol Medical School, University of Bristol, Bristol BS2 89HW, UK; 2Cardiac Surgery, University Hospitals Bristol, NHS Foundation Trust, Bristol BS2 8HW, UK; 3National Heart and Lung Institute, Imperial College London, London W12 0NN, UK

**Keywords:** congenital heart disease, children, tissue engineering, scaffolds, grow, fixation, decellularisation, acellular, cellular, biofabrication

## Abstract

Congenital heart disease (CHD) is the most predominant birth defect and can require several invasive surgeries throughout childhood. The absence of materials with growth and remodelling potential is a limitation of currently used prosthetics in cardiovascular surgery, as well as their susceptibility to calcification. The field of tissue engineering has emerged as a regenerative medicine approach aiming to develop durable scaffolds possessing the ability to grow and remodel upon implantation into the defective hearts of babies and children with CHD. Though tissue engineering has produced several synthetic scaffolds, most of them failed to be successfully translated in this life-endangering clinical scenario, and currently, biological scaffolds are the most extensively used. This review aims to thoroughly summarise the existing biological scaffolds for the treatment of paediatric CHD, categorised as homografts and xenografts, and present the preclinical and clinical studies. Fixation as well as techniques of decellularisation will be reported, highlighting the importance of these approaches for the successful implantation of biological scaffolds that avoid prosthetic rejection. Additionally, cardiac scaffolds for paediatric CHD can be implanted as acellular prostheses, or recellularised before implantation, and cellularisation techniques will be extensively discussed.

## 1. Introduction

Congenital heart disease (CHD) is the most prevalent birth defect globally [1]. Annually, this life-endangering disorder affects 1.35 million babies [2] and it is associated with an estimated 250,000 deaths worldwide [3]. Approximately a quarter of newborns with congenital cardiac abnormalities require invasive surgical intervention to replace defective or absent structures before their first birthday [4]. There are over 40 distinct subtypes of CHD and the incidence of some is increasing annually [5].

Early diagnosis and advances in cardiac surgery and interventional cardiology have significantly increased the survival of paediatric patients with CHD over the past several decades [6,7]. This is reflected in CHD demographics as the number of adults living with CHD now outnumbers the paediatric CHD population [8].

Despite these remarkable improvements, many interventions in CHD remain palliative, such as the Norwood procedure and overall Fontan palliation of single ventricle physiology, or coarctation of the aorta palliation [9,10]. A palliative intervention aims to improve the function of an abnormal heart, e.g., controlling heart failure and preparing for a later correction when the paediatric patient grows to a more suitably stable condition [10]. These interventions are usually performed to minimise symptoms, with patients experiencing cardiovascular complications over the long term, with significant residual haemodynamic or electrical conduction abnormalities [11,12].

Surgical replacement options for CHD patients include the use of valve substitutes, patches, and conduits. For instance, heart valve failure requires surgical replacement in 70% of cases [13]. These options, such as mechanical prostheses, have been demonstrated to be undeniably valuable in terms of preserving patients’ life. However, they have also been shown to negatively impact the quality of life, e.g., because of the necessity for anticoagulation therapy, which can hamper the active lifestyle of young adult patients with CHD, or issues related to calcification [14,15].

In addition, CHD patients frequently need multiple open-heart replacements of failing valves and/or conduits, thus being exposed to the risks of additional surgical interventions [16,17]. Not only do complex open-heart surgeries yield high short-term risk for neonates, but, in the paediatric population, additional long-term risks are associated with the inability of the prosthetic material to follow somatic growth, indeed leading to failure and re-operations [16,18,19].

The necessity to overcome such limitations of current clinically available prostheses for the treatment of paediatric CHD has prompted significant research into the development of novel bio-scaffolds. The search for readily available and biocompatible replacement parts endowed with growth and adaptive remodelling capacity, as well as durability over the patient’s lifetime, is ongoing in this field of paediatric research.

A review of the technical and bioengineering aspects relating to the fabrication of biological scaffolds for application in paediatric CHD is not presently available, despite this being a rapidly evolving area of research. This review, therefore, aims to provide a comprehensive overview of the techniques used to produce biological scaffolds for paediatric patients with CHD, discussing future tissue engineering (TE) based approaches to treat one of the fastest-growing populations in cardiology (Figure 1).

## 2. Tissue Engineering Approaches

Since its beginnings in the early 1990s [20], TE has gained much attention due to the immense potential of this interdisciplinary approach, becoming the subject of numerous biomedical applications. It combines molecular and cell biology with material science and chemistry, spanning the fields of engineering and medicine with the aim of generating biological surrogates as an answer to congenital tissue abnormalities or acquired damage [21]. In the context of producing scaffolds for applications in paediatric CHD, the intention of current TE-based research is to produce a living graft with native tissue function, including growth, remodelling, and regenerative capability [5,22,23]. TE has the potential to provide an improved alternative to existing treatments in cardiac reconstructive intervention, and strategies can utilise in vitro, in vivo, and/or in situ techniques [4,5,22,24].

Scaffold choice is crucial in TE, being a platform onto which cells are deposited and guiding tissue formation. An optimal scaffold mimics both the structural and mechanical properties of the native tissue, as well as supporting cell viability and growth [25]. Scaffolds used in TE fall into two categories, namely synthetic and biological. Synthetic scaffolds do not possess the native organic components of an extracellular matrix (ECM), reducing the likelihood of endogenous cell penetration and recellularisation of the scaffold. In this review, we focus on the application of biological scaffolds in CHD, encompassing tissue from human (homografts) and animal (xenografts) origin.

## 3. Production of Biological Scaffolds

Biological scaffolds are ECM biomaterials that have been shown to facilitate the constructive remodelling of many different tissues in both preclinical animal studies and in human clinical applications [26,27,28]. The ECM composing the biological scaffolds consists of the structural and functional molecules secreted by the resident cells of each tissue. Those molecules are mainly collagen (type I–VII) [29,30], glycosaminoglycans (GAGs) (including heparin, heparan sulfate, chondroitin sulfate, and hyaluronic acid) [31,32], fibronectin, laminin [33], and various growth factors such as fibroblast growth factor (b-FGF) and vascular endothelial growth factor (VEGF) [31,34,35,36]. They are arranged in unique three-dimensional (3D) patterns with specific compositions and distributions of the ECM constituents that vary depending on the tissue source. Typically, such scaffold materials are biodegradable unless processed in a manner such that irreversible crosslinks are created between the resident molecules.

Xenogeneic and allogeneic biological scaffolds have made major contributions to the field of cardiac TE. Tissue-engineered biological structures can be fabricated in vitro, where cells are cultured onto the biological scaffold by seeding, injection, or perfusion prior to surgery [37,38]. They can also be fabricated in situ, where remodelling occurs with the assistance of the host cell population within the patient’s body. In this case, off-the-shelf products could be used instantly during surgery, reducing the time needed to culture cells before implantation [39].

Although many natural biomaterials have reached the clinical trial phase, including alginate and collagen, decellularised ECM (dECM) biomaterials have unique structural characteristics which distinguish them from other biomaterials. These properties are necessary for the regeneration, repair, and remodelling of the defective regions in the heart. There are solid and soluble dECM biomaterials. Solid scaffolds can be used in their native structural form or as patches consisting of dry or hydrated sheets. Soluble dECM biomaterials are used as injectable hydrogels [40] and their main biofabrication tools are electrospinning and 3D bioprinting [41,42,43].

Preclinical studies in this area are numerous and focus on identifying alternative biological scaffolds possessing traits eluding current surgical options for CHD, including availability, growth capacity, and longevity matching the patient’s lifespan. Clinical use of commercially available ECM biological scaffolds for paediatric CHD aims to exploit the optimal structural and mechanical properties to achieve growth and biocompatibility [44]. In fact, the native ECM provides an ideal microenvironment for cells, supporting homeostasis, cell infiltration, and regeneration [44].

Processing of biological scaffold materials to mask or remove foreign antigens is an essential initial step of the fabrication process prior to any clinical application, including the treatment of paediatric CHD. Multiple immunogenic antigens exist on foreign tissue from allogeneic or xenogeneic sources. For example, N-glycolylneuraminic acid (Neu5Gc) is a sialic acid molecule present in most mammals, including cows and pigs, but is not found in humans [45]. As such, Neu5Gc is thought to induce calcium-mediated deterioration of implanted xenografts in a clinical cardiovascular setting [46]. Unlike Neu5Gc, major histocompatibility complex (MHC) class I and II antigens are present in human cells. This does not, however, prevent foreign MHC proteins on transplanted tissue causing a severe downstream T-cell and natural killer cell response and subsequent immune rejection [46].

Arguably, the most important xenoantigen discussed in the literature is Galα1,3Galβ1,4GlcNAc-R (α-Gal). The α-Gal epitope is a major xenoantigen responsible for hyperacute organ rejection from α-Gal donors to humans and is a complex barrier to trans-species implantation [23,47]. This carbohydrate antigen is produced by glycoproteins and glycolipids [48]. α-Gal is displayed in prosimians, New World Monkeys, and non-primate mammals, but is absent in apes, Old World Monkeys, and, importantly, humans [48]. It is known that α-Gal is synthesised by the glycosylation enzyme α1,3 galactosyltransferases (α1,3GT), but point mutations in the human sequence on chromosome 9 caused a frame-shift mutation and premature stop codon, inactivating the gene [48]. Resultantly, the α-Gal epitope was eliminated, and humans lost immune tolerance [48]. At this point in evolution, humans developed the ability to produce the anti-α-Gal antibody as the immunoglobulin (Ig) G, IgM, and IgA isotypes, most likely to combat α-Gal expressing pathogens [48]. The interaction between human anti-α-Gal antibody and the abundant α-Gal surface epitope on porcine endothelial cells is a significant obstacle to successful pig-to-human xenotransplantation [47,48]. The high-affinity binding of the α-Gal epitope and anti-α-Gal antibody activates a complement cascade, producing chemotactic cleavage complement peptides [48]. This results in substantial macrophage migration and a rapid immune response leading to rejection of the transplanted tissue [48]. In addition to hyperacute rejection, complement activation, and acute vascular rejection, the T-cell-dependent response to α-Gal epitopes induces chronic inflammation, leading to immune-mediated xenograft rejection [47]. To combat this barrier to xenotransplantation, α-Gal must not be exposed to the anti-α-Gal antibody. As such, it is important to remove or mask the α-Gal epitopes on α-Gal donor tissue to circumvent rejection in xenotransplantation. Two methods are commonly used to achieve this: fixation and decellularisation.

Next, we discuss the processing of biological scaffolds by chemical fixation or decellularisation to reduce antigenicity.

### 3.1. Chemical Fixation of Biological Scaffolds

In order to reduce their antigenicity, biological grafts can be treated chemically. The principle behind the fixation of tissue is not to remove the cellular component, but to kill the cells and stabilise the structures by chemical crosslinking of cellular proteins [49].

Low concentrations of aldehyde fixatives are the most widely used, for example, glutaraldehyde (GA) [46]. GA creates chemical bonds crosslinking ECM components, increasing mechanical resistance and tissue stability, as well as maintaining tissue sterility and prolonging potential storage time [48,50].

GA-fixed animal pericardium, usually of bovine or porcine origin, is currently used as an off-the-shelf commercially available cardiac valved xenograft source for the replacement of defective structures in CHD patients [51,52]. Though the fixation process preserves tissue stability and sterility, and masks antigenicity, the protocols do not remove major native xenoantigens, such as α-Gal, and toxic GA remnants remain in the graft [52]. As such, these implants have known mid- and long-term complications associated with severe calcification due to immunological incompatibility and resulting in chronic inflammation [52]. For cardiac valve replacement, leaflet structural deterioration is the primary cause of mid/long-term prosthetic valve failure, particularly in the paediatric population [52]. These GA-fixed xenografts have been shown to possess limited in vivo recellularisation potential and are, therefore, unable to remodel or grow with the patient, as is necessary for paediatric CHD patients [23,51]. Subsequent mechanical failure of the xenograft necessitates surgical replacement [52]. As such, despite offering life-saving temporary solutions, applications to paediatric CHD are imperfect.

### 3.2. Decellularisation of Biological Scaffolds

As an alternative approach to increase the biocompatibility of xenografts for use in CHD, detergent and/or enzyme treatments have been optimised with the aim to decellularise the structure and therefore reduce antigenicity [53,54,55,56,57,58,59,60,61,62,63,64,65,66,67,68,69,70].

Decellularisation of biological tissues prior to transplant into the cardiac environment has the potential to surpass previously used fixation techniques, allowing for enhanced biocompatibility via complete removal of xenoantigens, the absence of GA fixative traces, ECM conservation, and subsequent in vitro or in vivo recellularisation [52,71]. This would facilitate the restoration of tissue viability and function, including growth and repair [51]. Despite this, conditions required to achieve decellularisation inevitably reduce tissue integrity [51]. As such, much research has gone into the development and optimisation of novel decellularisation techniques to maximise biocompatibility and enhance the immunotolerance of the biological scaffold.

Decellularisation techniques for biological tissue can be broadly categorised into three groups: chemical, enzymatic, and physical, with the best results often achieved using a combination of approaches [46]. At their core, the purpose of all decellularisation methods is to eliminate native cellular and genetic material, including cell membrane phospholipids, mitochondria, and nucleic acids, whilst preserving the ECM ultrastructure composition and mechanical properties, therefore function [44,46]. If successful, this will diminish antigenicity and retain the mechanical integrity and signalling molecules of the natural tissue matrix, allowing endogenous cell recruitment post-implantation [5,23].

Achieving the perfect balance of full decellularisation while maintaining mechanical properties is challenging, demanding a multifaceted approach optimised to the specific material and clinical application. To date, three key criteria defined by Crapo et al. [72] are commonly used for newly developed protocols: (i) <50 ng dsDNA per milligram dry weight ECM, (ii) <200 base pair fragment length, and (iii) lack of visible nuclear material in histologically stained tissue sections [72]. However, these criteria alone are insufficient to objectively and quantitatively evaluate decellularisation success. For example, detergents used in chemical decellularisation cause cytotoxicity if not fully removed, a factor not considered in the above criteria [44]. The universal aim is to obtain a biocompatible scaffold, but legal criteria defining what constitutes an acellular graft have not yet been established [44].

An early report documenting the disastrous clinical results following the implant of SynerGraft^®^ decellularised porcine pulmonary valves is a cautionary tale against the application of incompletely decellularised xenografts in paediatric CHD patients [73]. SynerGraft^®^ processing is a patented decellularisation technology used on the tissue of both animal and human origin to produce acellular valve replacements with endogenous repopulation capacity [74,75]. The SynerGraft^®^ decellularised porcine heart valve was introduced as the earliest tissue-engineered alternative to conventional xenograft valves in Europe. In 2002, four children received model 500 (n = 2) or model 700 (n = 2) SynerGraft^®^ valves [73]. Sadly, three children died, one due to rapid valve rupture a week postoperatively, and two due to critical graft structural degeneration of the valve leaflets and wall at 6 or 52 weeks after implant. The fourth child underwent a prophylactic explant 2 days after implantation, and severe inflammation could already be observed on the outside of the explant. All explants revealed calcification and early explants exhibited an acute neutrophil granulocyte and macrophage reaction. This initial non-specific inflammatory response was followed by a strong lymphocytic reaction, as was seen in the histological analysis of the 1-year explant. Unsurprisingly, no endogenous recellularisation of the porcine tissue was observed. Importantly, calcific deposits were found on pre-implant SynerGraft^®^ samples and only partial decellularisation of the porcine collagen matrix as shown in [73]. This case clearly illustrates the risk of immunogenicity and consequent degeneration associated with xenografts in paediatric CHD treatment, warning against the application of poorly decellularised animal tissue.

All methods described in the literature are imperfect. Alterations to ECM composition, integrity, and organisation is inescapable to some measure, but the extent of disruption to macromolecules, such as proteins, GAGs, and collagens, is dependent on numerous interlinking factors. Thus far, achieving 100% cell removal with absolute preservation of ECM ultrastructure composition and macromolecular properties has not been reported.

Chemical decellularisation techniques include the use of surfactants, such as the detergents sodium dodecyl sulphate (SDS) [23,53,54,55,58,59,63,64,65,66], sodium deoxycholate (SD) [59,60], and Triton X-100 [58,59,60,61,62,63,64], which lyse cells by disrupting the phospholipid bilayer of the cell membrane. As an alternative to surfactants, acids and bases including sodium hydroxide (NaOH) can be used to solubilise cell membranes and nuclear material [56]. Enzymatic decellularisation, on the other hand, utilises biological agents to break down nucleic acids and proteins. DNase [53,59,60,61,63,65,66] and trypsin in combination with the chelating agent EDTA [54,59,60,61,62,63,65] are commonly used with chemicals. Nucleases digest nucleic acids into shorter chains, assisting their removal, while EDTA sequesters metal ions, disrupting cell-ECM attachment [5]. Finally, physical decellularisation methods include mechanical agitation to improve lysis efficiency and debris removal, and supercritical carbon dioxide (scCO2) [57]. More details on these methods can be found in [46].

## 4. Biological Scaffolds for CHD

Once successfully fixed or decellularised, biological scaffolds for CHD can be used as acellular prostheses, or as a platform for cell seeding prior to implantation. We will review both approaches in specific CHD applications, discussing both acellular and recellularised biological scaffolds. Table 1 summarises the advantages and disadvantages of biological scaffolds from human and animal sources, as well as their use as acellular or recellularised prostheses.

### 4.1. Acellular Scaffolds for CHD

Biological bioactive materials have structural properties that are necessary to support cell health, function, and tissue repair through the presence of a pool of growth factors, matricellular proteins, and complex ultrastructural compositions [43]. Even after the in vivo remodelling of the acellular scaffold’s ECM inside the patient, the degradation products have been shown to influence endogenous cell activity [76]. These may play a major role in tackling the limited regenerative capacity of the heart [76]. More evidence proves that the plasticity and regenerative capacity of endogenous cells and the anatomic structure of the heart are the main reasons behind tissue remodelling induction and functional improvement proving the potential of using scaffolds that exploits either fixed ECM or dECM [76]. The acellular scaffolds could be used as a stand-alone therapy or as a vehicle used to deliver therapeutic agents to the heart. These can specifically help in tackling issues related to cell survival and engraftment that occur due to the hostile nature of the damaged heart [43,77]. All in all, this helps in pro-regenerative signalling which ultimately leads to cardiac tissue repair and remodelling. This is not possible if acellular scaffolds are immunogenic, unable to maintain cellular attachment and proliferation, or affect immunophenotype or differentiation capabilities. These scaffolds must have similar biomechanical properties, biodegradation rate, permeability, and flexibility as the native cardiovascular tissue; however, they exhibit uncontrollable degradation and inadequate mechanical properties.

Acellular scaffolds could be used in their natural structure (i.e., whole hearts [78]), patches (i.e., sections from whole hearts [79]), injectable form (i.e., crosslinked heart-tissue-derived ECM gel [80]), 3D bioprinted constructs (i.e., complex, biomimetic 3D bioprinted vascular structures composed of ECM hydrogel sourced from omental tissue [81]) and 3D electrospun scaffolds (i.e., composed of decellularised porcine cardiac tissue blended with poly (ethylene oxide) [16]). Examples of such scaffolds from xenogeneic and allogeneic sources include porcine small intestinal submucosa (SIS), human amniotic membrane (AM), and porcine urinary bladder matrix (UBM), briefly discussed below. Next, we discuss acellular scaffolds classified by origin, being homografts or xenografts.

#### 4.1.1. Homografts for CHD

Biological substitutes are either homografts or of animal origin [13,82]. Homografts can be divided into two classes: autografts, i.e., tissue from the patient’s body, or allografts, i.e., tissue from a human donor (Figure 2A). Autografts and allografts possess favourable traits, including suitable haemodynamic performance, manipulability, and capacity for growth and repair owing to cellular infiltration [83]. Though autografts are not vulnerable to immune rejection, allografts are sourced from a donor patient or cadaver, and can therefore elicit an immune response. Therefore, allografts need to undergo decellularization or fixarion (Figure 2B,C) before further implantation in patients (Figure 2D).

Different sources of homografts include tissue obtained from the pericardium, decellularised cardiac tissues, AM, and ECM [84]. Cardiovascular dECM such as pericardium, myocardium, and pulmonary artery tissue could be used as potential sources for cardiovascular therapeutic applications in paediatric CHD. Site-specific dECM scaffolds exhibit the same composition and spatial arrangement of ECM constituents as the target tissue, allowing adequate cell–cell interactions to occur and paving the way towards tissue morphogenesis and thus repair and regeneration of the heart [85,86].

A widely used autologous biological graft in CHD is represented by the pericardium, which is commonly used for surgical correction of the pulmonary valve. Despite the diffused application, the autologous pericardium has shown a tendency to degenerate due to endothelial stress caused by blood flow through the artery [87]. Another example of the use of autografts in the treatment of paediatric CHD is the Ross procedure, which involves transplanting the native pulmonary valve into the aortic position, therefore using a pulmonary autograft to replace the defective aortic valve [88,89,90,91,92,93]. The Ross aortic valve replacement strategy enables the growth of the valve implanted in the aortic position and is therefore favourable for paediatric aortic valve replacement. However, this does not obviate the need for a replacement valve from a non-autologous origin since the pulmonary autograft must then be replaced. Additionally, the Ross procedure is limited to CHD where the aortic valve is defective, and not a solution where the pulmonary valve is the prime cause of the diagnosis.

To replace the pulmonary or aortic valve, decellularised allogeneic heart valve tissue has been used in CHD treatment. The durability of these acellular scaffolds is dependent on endogenous cell recruitment, including smooth muscle cells (SMCs) and fibroblasts, which can lay down ECM. In a retrospective 10-year comparison of SynerGraft^®^ pulmonary allografts (n = 163, mean age 17.3 years, range 2 days to 74 years) and standard cryopreserved pulmonary allografts (n = 124, mean age 12.6, range 3 days to 56 years) used in right ventricular outflow tract (RVOT) reconstruction across three centres, the SynerGraft^®^ group had significantly a higher absence of conduit dysfunction (83% vs. 58%), suggesting that the SynerGraft^®^ decellularisation technology decreases allograft immunogenicity, increasing longevity of the conduit [74]. In 2019, a study of 364 explanted decellularised acellular allogeneic heart valves (pulmonary n = 236, aortic n = 128) showed considerable non-inflammatory recellularisation, demonstrating that in vivo recellularisation to realise long-term functionality of the implant is possible. This promising outcome, however, is overshadowed by the data from allogeneic heart valves explanted after less than one year when applying the results to paediatric CHD. These data show cardiac grafts were not uniformly repopulated with host cells, demonstrating recellularisation by endogenous cells is slow in comparison to the rapidly growing heart of a child, requiring multiple months [94].

dECM AM is a widely used biocompatible scaffold material for CHD as it has excellent biological properties that promote wound healing and tissue regeneration through mechanisms such as angiogenic, anti-inflammatory, antifibrotic, and anti-bacterial activity [95,96]. This is due to the presence of the growth factors, structural proteins, and glycoproteins necessary for wound healing, tissue remodelling, and regeneration [97]. dECM AM showed high biocompatibility in vitro with mesenchymal stem cells (MSCs) remaining metabolically active after 72 h of culture on the membrane [97]. However, the drawbacks of using dECM AM include differences between donors, limited biomechanical properties and biodegradation rates, changes in the membrane properties due to the preservation or decellularisation techniques, and risk of disease transmission [95].

A vascular graft produced from a sacrificial fibrin gel with encapsulated donor fibroblasts remodelled when placed in a bioreactor. Following this, the ECM was decellularised to create an off-the-shelf ECM product and implanted as a pulmonary artery replacement in three lambs [98]. This proof-of-concept model showed growth potential (increased body weight by 366%) coupled with an increase in graft diameter (56%) and volume (216%). Furthermore, the explanted grafts exhibited physiological strength and stiffness coupled with full lumen endothelialisation and infiltration of mature SMCs. The grafts showed no evidence of calcification, aneurysm, or stenosis, and exhibited significant elastin deposition coupled with enhanced collagen content (465%).

In a transcatheter aortic valve replacement animal study, tissue-engineered heart valves (TEHV) were fabricated with the use of a stent and human-cell-derived ECM [99]. Results showed that the TEHV did not collapse in the aortic environment and displayed normal leaflet motion, lack of stenosis or paravalvular leak, and free coronary flow. Importantly, the TEHV showed intact structural integrity and clear signs of in situ host cell repopulation.

A 10-year follow-up of fresh acellular decellularised pulmonary homografts for pulmonary valve replacements in 131 patients compared these with cryopreserved pulmonary homografts and bovine jugular vein conduits [100]. Data showed the decellularised pulmonary valves were safe to use with no signs of endocarditis, non-repeated operations, good functionality with mild regurgitation, and absence of stenosis [100].

In summary, results as to whether in vitro or in situ recellularisation is better are conflicting [101]. When considering paediatric CHD surgery, insufficient spontaneous host recellularisation highlights the potential of, and perhaps the need for, in vitro recellularised scaffolds. Demand for homograft replacements, however, far outweighs supply, especially in paediatric CHD. As such, alternative tissue sources must be considered.

#### 4.1.2. Xenografts for CHD

Homograft represents the preferred option for CHD surgical replacement due to the more accurate matching of native tissue, and because they are considered more durable and less immunogenic than xenografts. Nevertheless, their availability is inadequate for clinical CHD scenarios, particularly in the context of small-valved conduits for neonates and infants. This represents a significant challenge in terms of use in paediatric CHD [5]. Xenograft substitutes, most commonly from bovine or porcine sources, represent alternative biological scaffolds for cardiac surgery [13,82] (Figure 2E). Xenografts need to undergo decellularization or fixarion (Figure 2F,G) before further implantation in patients (Figure 2H) to avoid immune response.

Xenografts of porcine origin, include Matrix P valves (MPV), Matrix P plus^®^ valves (MP+V), urinary bladder matrix (UBM), Proxicor^®^, and dECM hydrogels.

MPV and the MP+V, comprising decellularised porcine valves with a GA-fixed equine conduit component, are commercially available xenogeneic heart valves used in CHD treatment [102]. An analysis of 17 MPV and 10 MP+V pulmonary valves implanted into patients of different ages (10 months to 39 years) revealed a high occurrence of valve replacement (52%), fibrosis, and severe inflammation on histological assessment, with no evidence of endothelialisation of any valve [103]. Magnetic resonance imaging supported the conclusion that MPV and MP+V failure can be attributed to immune-mediated graft degeneration [103]. A later retrospective study comparing 18 implanted MP+V xenografts (3 valve failures after 25.3 ± 14.6 months) and 14 decellularised allograft valves (no valve failures) established levels of tissue-specific antibodies following grafting, linking these to the occurrence of valvular insufficiency [102]. Patients receiving MP+V xenografts had significantly increased IgG levels towards both the decellularised and GA-fixed tissue compared to allogeneic valves. The decellularised component of MP+V, however, elicited a lower antibody generation than the GA-fixed component. Valve failure was not linked with significantly higher IgG generation. Tissue-specific antibody production normalised to the control baseline after 12 months in the decellularised allograft patients. This study confirmed a more potent immune response to MP+V than decellularised allogeneic counterparts. It also concluded that increased antibody levels are raised in response to the GA-fixed conduit component of MP+V xenografts compared to their decellularised valve leaflets and even more so compared to decellularised valves of allogeneic origin. Overall, the failure of these cardiac substitutes has been connected to graft immunogenicity and host antibody generation, but decellularised allograft valves do not exhibit the same level of failure [102].

A preclinical trial assessed non-GA fixed, SDS decellularised porcine TEHV for use in trans-species implantation, with the view to apply these in CHD treatment in the future [23]. Here, SDS decellularised porcine aortic valves, sterilised with scCO2, were implanted into the RVOT of juvenile sheep (n = 5) for 5 months [23]. This trial demonstrated recellularisation by host cells and appropriate valve haemodynamics for the 5-month period. Following the explant, myofibroblast-like cell infiltration was shown, as well as deposition of collagen fibrils and formation of an endothelial layer on the explant surface. Valve cusps increased in tensile stiffness and maintained their strength, confirming recellularisation by collagen synthesising cells.

dECM UBM is a biological scaffold harvested from either human or, more frequently, porcine sources. Initially, it is created from the removal of the bladders from the body where it is then mechanically delaminated, and the detrusor muscle is segregated from the lamina propria and urothelium. The lamina propria is then decellularised and used as various end products such as patches, injectable particulate suspensions, and hydrogels [104]. Remlinger et al. reported the comparison in efficacy between dECM UBM patch and dECM organ-specific cardiac patch in the repair and remodelling of full-thickness defects in the RVOT of rats [105]. Results showed both patches facilitated cardiac function and cellular infiltration within the RVOT. However, the dECM UBM patch outperformed the dECM cardiac patch in terms of remodelling, including complete degradation of the patch before the formation of new tissue, faster cellular infiltration, and the presence of cardiomyocytes with evident sarcomeric morphology.

ProxiCor^®^ (previously known under the commercial name of CorMatrix^®^) is a commercially available decellularised porcine SIS scaffold [2,106,107,108]. It has gained great attention in the cardiovascular field [109] since 2010 due to its ability to support the ingrowth of host cells, promote cell proliferation and differentiation, contractility, absorbability, neovascularisation, and a lack of immunogenicity [107]. Furthermore, it can grow, repair, and remodel congenital heart and vascular defects [110,111], facilitate pericardium reconstruction [112], valve reconstruction in adults and children [113], endocardium reconstruction, and the repair of post-myocardial infarction defects [114] through the promotion of native tissue growth without the need for repeated surgeries [115]. Despite the advantages of SIS-dECM, drawbacks in the long term include reactive inflammation, calcification, infections, adhesion, and local stiffness at the site of injury [116].

A SIS-dECM patch, used to close ventricular and atrial septal defects in 73 paediatric CHD patients (average age 22 weeks) [117], showed lower rates of reoperations compared to other patch materials. Additionally, it exhibited more flexibility and ease of use. Nelson et al. reported a prospective study based on explanted acellular Proxicor^®^ scaffolds from infants with CHD [118]. Assessments based on haematoxylin and eosin, Movat pentachrome, and Masson’s trichrome stains showed that all explanted patches exhibited chronic inflammation, fibrosis, and foreign body giant cell reaction. Furthermore, the explanted intracardiac patches displayed calcification, elastic fibres, eosinophils, no infiltration of host cells, and inhibited remodelling of the cardiac tissue after 21 months of implantation. In another study, four infants suffering from CHD valvular defects received personalised porcine SIS tri-leaflet valves [119]. A short-term clinical follow-up based on haemodynamic assessment (TEHV scaffolds vs. standard porcine bioprosthetic valves) revealed comparable flow and pressure profiles, despite significantly higher forward flow energy losses exhibited by the engineered valves. This could be explained by the stiff nature of SIS material versus GA-fixed control material [120].

dECM in the form of an injectable hydrogel scaffold is an alternative approach for use of biological scaffolds in CHD. Wu et al. reported the fabrication of TEHV through the encapsulation of NIH 3T3 fibroblasts in injectable hydrogels derived from decellularised porcine mitral valve chordae, decellularised aortic valves, and decellularised mitral valve leaflets, remarking on their potential use in TEHV due to their ability to grow and remodel with the somatic growth of the patient [121]. This proof-of-concept study showed differences in collagen concentration and similarities in the GAG content, nanofibrous structures, and gelation kinetics between the three hydrogels. Furthermore, the NIH 3T3 fibroblasts encapsulated in the injectable hydrogels maintained cell survival up to 7 days.

Bovine sources of xenografts for use in CHD include Contegra^®^ and CardioCel. The Contegra^®^ xenograft is an integrated valved conduit composed of the bovine jugular vein and is currently the surgical standard for RVOT substitution in paediatric CHD patients [122]. Since its development in 1999, Contegra^®^ has been widely adopted by surgeons due to its availability, size range, low reported calcification incidence, and comparatively low cost [123]. Studies have reported favourably on the outcome of Contegra^®^ implantation in comparison to homografts or porcine xenografts [124,125,126]. Despite this, distal stenosis is an issue, with one study reporting 51% severe distal stenosis after 2 years in Contegra^®^ recipients [127]. Supravalvular stenosis has also been reported [128].

Tissue-engineered bovine pericardium (CardioCel) was implanted over a period of 24 months in 135 patients (neonates, infants, and children older than 365 days) suffering from various congenital heart defects such as septal defects, repair of pulmonary arteries, intra-atrial/intraventricular baffles, repair of systemic arteries, valve repair, repair of systemic veins, and Fontan procedure [120]. Results showed no patients experienced any signs of calcification at 36 months. Furthermore, CardioCel displayed good haemodynamic performance with comparable results when placed in systemic and pulmonary circulations.

Despite the diffused application of xenografts for CHD that has led to many commercially available products, animal-derived products still have limitations. In fact, clinical data show that long-term durability of the GA-fixed graft is limited by severe dystrophic calcification due to the permanence of fixative remnants having potent cytotoxic effects [129]. In addition, acellular prostheses lack growth potential [22], which is essential in paediatric CHD [23]. Not only is calcification over time a major problem for long-term durability, but these prostheses also suffer from poor patency and a high rate of thrombotic occlusions [24]. This is particularly problematic for small-diameter vascular grafts in newborns [82].

Another strategy to produce tissue-engineered biological scaffolds for paediatric CHD is an in vivo method that employs the innate inflammatory response as a natural bioreactor. A non-degradable mould is implanted subcutaneously, encapsulated with deposited ECM, including collagen, elastin, and proteoglycans, and then explanted [5]. The produced ECM is physically removed from the non-degradable mould, leaving just the biological ECM-based scaffold. These grafts can be repopulated with cells prior to implantation or used as cell-free off-the-shelf prostheses for in vivo cellularisation. The graft can be implanted back into the original host as an autograft (Figure 2I), as has been previously carried out with BIOTUBE vascular grafts [130], or applied as an allograft [131].

A Japanese group used this method to produce pulmonary valved conduits via in vivo encapsulation of silicone rods implanted in the dorsal subcutaneous space of beagles for 4 weeks [132]. The deposited connective tissue was removed from the silicone, and the tri-leaflet structure was implanted as an allogeneic valved conduit in the pulmonary position for 84 days (n = 3). Unexpectedly, the allografts were not rejected by the host, though the reason for this was unclear. Surface endothelialisation and formation of neointima with an elastin network were observed in explants. The group suggest implanting the connective tissue as an autograft in the future to investigate the rejection response. Though ECM-based grafts hold potential in terms of diminished immunogenicity, this resource-intensive method necessitates three surgical interventions: subcutaneous mould implantation, explant of the encapsulated mould, and implantation of the neotissue. Additionally, few in vivo studies have taken place and long-term observations are lacking. At present, percutaneous implantation of a non-biodegradable mould to generate an ECM-based graft has only been carried out in animals, producing a xenograft of either autologous or allogeneic origin. However, this technique could theoretically be extended to humans in the future, creating autologous homografts.

### 4.2. Cellular Scaffolds for CHD

As mentioned, despite fixed and decellularised xenografts having the advantage of accessibility, lack of growth potential, limited remodelling capacity, absence of repair, and poor longevity due to their acellular nature are well-known complications, and particularly prominent in the paediatric CHD population [22,23,24,82,133]. The use of suboptimal prostheses for young patients with CHD often results in a series of high-risk open-heart operations [133]. In vitro cellularisation of an acellular biological scaffold can be performed when the basal properties of the ECM are not sufficient for the clinical CHD application and hold the potential to circumvent limitations associated with acellular prostheses. Here, we describe the main cellularisation methodologies that have been explored for biological scaffolds for CHD, categorised according to the method in which cells are seeded.

Homeostatic maintenance of the network of interwoven signalling pathways in a living structure requires not only a scaffold, but also cellular crosstalk. Therefore, to facilitate the recapitulation of native-like properties, the second component of a tissue-engineered product is cellularisation of the prepared scaffold framework. Without a cellular component, somatic growth capacity, repair, and adaption are impossible [5]. The requirement for these features is heightened when considering life-long paediatric CHD treatment. As mentioned, cellularisation of biological scaffolds can rely on the body as a bioreactor to repopulate an acellular implant. However, in vivo clinical data indicate the rate of spontaneous recellularisation as too slow in comparison to the growth requirements of the scaffold in a paediatric setting [94]. As such, the future of paediatric cardiac prostheses is pre-implantation in vitro cellularisation.

In vitro seeding of the scaffold with autologous or allogeneic cells prior to implantation can facilitate the maturation of the cell-laden product. Biological scaffolds that have gone through the decellularisation process to avoid immunogenic rejection often require recellularisation to achieve the reconstruction of blood vessels, in which the presence of an endothelial lining in the lumen is crucial to avoid thrombogenic events. In other cases, recellularisation with autologous cells is needed to provide the capability to grow, remodel, and release new ECM proteins to improve mechanical properties. In vitro seeding can be achieved by manual cell deposition, injecting cells onto a scaffold, or 3D bioprinting (Table 2).

The use of autologous cells in a personalised medicine approach will support immunocompatibility [5]. Desirably for paediatric CHD patients, the cell population for seeding should be readily accessible at the time of birth [82]. Whereas primary cell types have a limited lifespan, stem cells are self-renewable, meaning their utility could endow the graft with growth potential [134]. MSCs are an immature cell type with multilineage differentiation potential, high proliferative capacity, and power to induce immunotolerance, and have therefore been an attractive choice in cardiac regenerative medicine [134,135]. MSCs can be isolated from multiple sources, many of which require invasive harvesting, for example, bone marrow MSCs [136]. Alternative MSC sources include the umbilical cord; for example, the umbilical cord is a potential source of allogeneic human umbilical cord blood MSCs (hUCB-MSCs), perivascular tissue-derived MSCs, or Wharton’s jelly MSCs when the prenatal diagnosis of the defect is made [133,137,138]. In this way, tissue usually discarded can be used to harvest MSCs immediately after birth in a non-invasive manner. Our group has validated the use of hUCB-MSCs for TE, proving high-quality MSCs can be expanded to clinically useful quantities from this source, with maintained MSC phenotype and cells retaining multipotency in vitro until stimulated to differentiate [133]. The application of allogeneic or autologous MSCs could generate a tissue-engineered structure able to repair, remodel, and grow concurrently with the surrounding host tissue.

#### 4.2.1. Manual Seeding

The method of static (manual) seeding is an inexpensive and simple technique for directly depositing cells onto a scaffold [139] (Figure 3A). Historically, the first clinical application of a cellularised graft for CHD was in 2006 by Cebotari et al. The study reported the use of decellularised pulmonary heart valve allografts reseeded with autologous peripheral mononuclear cells isolated from human blood in two paediatric patients suffering from congenital pulmonary valve failure [140]. Both patients experienced mild pulmonary regurgitation postoperatively. After a 3.5-year follow-up, results showed an enhancement in the valve annulus diameter, a reduction in valve regurgitation, a decrease (for patient A) and an increase (for patient B) of mean transvalvular gradient and constant or reduced right-ventricular end-diastolic diameter. Degeneration of the implanted valve was not detected in either patient. The ability to remodel and grow with the child showed the potential of the recellularisation approach and paved the way towards the use of cellularised grafts for CHD.

After this precursor study, the approach was widely explored and resulted in many in vitro and in vivo studies. A porcine dECM was used in an in vitro proof-of-concept model of engineered heart tissue. Human-induced pluripotent stem cell-derived cardiomyocytes and primary cardiac fibroblasts were seeded on the decellularised matrix and exhibited significant electrical and functional characteristics [141]. The engineered heart tissue facilitated electrical conduction when electrically paced up to 2 Hz frequency, showing the possible use in the treatment of single ventricle heart defects.

Other in vitro studies showed the use of biological grafts reinforced with synthetic material to improve the structural strength. Human umbilical artery-derived SMCs seeded on reinforced porcine blood-derived fibrinogen exhibited unique topographical and biochemical cues resulting in biologically inspired arrangement of ECM with an increased presence of elastic-fibre proteins [142]. In an in vitro model of CHD reconstruction, rat aortic endothelial cells, and rat bone marrow-derived macrophage were cocultured and seeded onto hybrid homogenised pericardium matrix and showed increased expression of healing cytokines [143].

Zhao et al. reported the use of novel polyurethane/SIS patches seeded with hypoxia pre-treated urine-derived stem cells in comparison to commercial bovine pericardium patches in the reconstruction of RVOT in 40 rabbits [144]. The stem cells seeded onto polyurethanes/SIS patch prevented fibrosis and stimulated vascularisation and muscularisation, yielding improved right heart function.

Recent studies described the use of progenitor cells for CHD applications, showing high performance in terms of adhesion, survival, and proliferation when seeded on decellularised porcine pulmonary artery [145] and cardiac dECM [146]. Importantly, the latter demonstrated the presence of tissue-specific ECM proteins stimulating the differentiation of cells into cardiomyocyte-like cells without the use of external induction factors [146]. In another study, a decellularised aortic heart valve demonstrated the ability to stimulate a significant increase in the recellularisation of the seeded endothelial progenitor cells in in vitro and in vivo environments, as well as decrease inflammation in a bioreactor when functionalised with stromal-derived factor-1α (SDF-1α) [147]. SDF-1α-decellularised aortic heart valves preconditioned in a bioreactor have the potential to be used as functional, robust heart valves that can grow and remodel.

Our group reported the use of decellularised swine SIS graft seeded with swine MSC-derived vascular SMCs for the replacement of pulmonary arteries in piglets [82]. After 6 months of surgery, the seeded grafts exhibited arteries with greater circumference compared to non-seeded grafts, without any indication of aneurism.

In the attempt of exploring a novel alternative for CHD corrective surgery, freeze-dried AM-based scaffolds were tested as vascular grafts. When implanted in the left pulmonary artery of piglets, the cell-seeded multiple-layered AM-based graft showed promising results in terms of suitability and biocompatibility for vascular repair due to the development of newly formed endothelium in the intima, a SMC-rich medial layer, and an adventitia containing new vasa vasorum [133].

Recently, a prosthetic graft composed of pericytes from neonatal swine was seeded onto Proxicor^®^ conduits to reconstruct the left pulmonary artery of 9-week-old piglets [2]. The in vitro results demonstrated the viability and incorporation of the cells in the outer surface of the conduit and suggested decreased stiffness of seeded conduits as opposed to non-seeded conduits. After 4 months in vivo, results showed the structural and functional incorporation of the grafts with the host tissues.

The manual seeding of MSCs derived from allogeneic Wharton’s jelly on SIS was compared to commercial unseeded SIS for the reconstruction of the main pulmonary artery in growing piglets [148]. After 6 months post-surgery, in vivo results suggested the incorporation of the seeded grafts with the native host environment and a significant growth potential of the seeded grafts.

An alternative model to the pig is the rodent model [149,150]. In one study, a commercial ultra-foam collagen sponge was seeded with MSCs from cyanotic CHD patients in comparison to MSCs from acyanotic CHD patients and used to reconstruct the RVOT of rats. Four weeks post-surgery, evaluations showed that the cytokine-treated collagen patches seeded with MSCs sourced from bone marrow mononuclear cells presented preserved morphology, increased cell survival, and elevated angiogenesis in contrast to MSCs from acyanotic CHD patients [149]. Saito et al. reported the development of scaffold-free arterial grafts consisting of 10-layered cell sheets of human umbilical arterial SMCs exposed to periodic hydrostatic pressure (one cell layer at a time) and implanted in the aorta of rats [150]. Results showed that the scaffold-free construct displayed patency and full endothelialisation according to echocardiographic and histological assessments.

The main drawbacks of this technology include inefficiency, impracticality compared to other technologies, and random deposition of cells resulting in a heterogenous distribution of cells across the tissue-engineered scaffold [151] (Table 3).

#### 4.2.2. Cell Injection

Cell injection is a simple technique which entails loading a cell suspension into a syringe and directly injecting it into a scaffold through a hollow needle, infiltrating deep into the scaffold [139] (Figure 3B).

Patel et al. reported the use of a two-stage cell seeding technique in addition to perfusion culture on a chitosan bioengineered open ventricle scaffold to fabricate a two-stage perfusion cultured ventricle to be potentially used in the treatment of hypoplastic left heart syndrome [152]. This was done by directly injecting primary rat neonatal cardiac cells into the scaffold and encapsulating it with a 3D fibrin gel artificial heart muscle patch, which was then perfused for 3 days in culture. Results revealed cell retention post-culture was 5%, and intercellular connections and gap junctions between deposited cardiac cells were formed on the scaffold surface. Two years later, the same group reported the fabrication of the novel bioengineered complete ventricle (BECV) and its potential use in the treatment of the left ventricle associated with paediatric hypoplastic left heart syndrome [153]. BECV is composed of bioengineered trileaflet valve (BETV) moulds coupled with chitosan scaffolds used to biomimic the human neonate aortic valve structure. BECV was fitted into a bioengineered open ventricle, which was cellularised using the two-stage cell seeding technique paired with a perfusion culture of rat neonatal cardiac fibroblasts for 3 days. Results showed the average pressure varied from 0.06 to 0.12 mm Hg. Additionally, there was evidence of syncytial-type cardiomyocyte aggregates and non-homogenous distribution of cardiac fibroblasts on the BECV surface.

The drawbacks of this technique include inconsistency of cell suspension across the layers of the scaffold and exposure of cells to high shear rate and pressure during the injection process. Additionally, this slow and time-consuming process is only efficient for small-scale seeding [139] (Table 3).

#### 4.2.3. 3D Bioprinting

The major challenge facing TE applied to paediatric CHD is the inability to fabricate functional constructs with full vascularisation and synchronous contractile activity. This limitation is due to the following factors; (i) partial tissue survival post-transplantation; (ii) generation of engineered tissue of clinically relevant sizes; (iii) optimising the mix of cardiac cell types; (iv) difficulty in sourcing cells from patients; (v) the immature phenotype of stem cell-derived cardiac cells; (vi) fate and safety concerns of potential undifferentiated stem cells; and (vii) immunogenicity which will require immunosuppression [154].

In the last decade, 3D bioprinting emerged as a potential approach to overcome these limitations. In fact, bioprinted structures have the capability to play a key role in supporting cells by mimicking the mechanical characteristics of native ECM through overall patch geometry, size, survival, and function (Figure 3C). Furthermore, hydrogel-based materials could also be used to biomimic the native ECM environment by creating cell binding sites to allow cellular functions, ease of access to nutrients, and paracrine signalling [97]. To be able to 3D bioprint structures composed of natural hydrogels (e.g., collagen, fibrin, dECM), solubilisation of natural tissues is necessary. The most widely used solubilisation technique in the literature is the enzymatic digestion technique which forms hydrogels with retained mechanical and cell-friendly physiological properties. This method uses pepsin dissolved in an acid solution to break down the ECM proteins into smaller peptides. Prior to that, the biomaterial should be in the form of a fine powder to allow maximum solubilisation of the tissue. Depending on the type of biomaterial, the solubilisation could take a few hours to several days to complete under constant shaking/stirring at specified temperatures. Following this, solutions are adjusted to neutral pH using varying concentrations of NaOH and phosphate buffer solution on ice to prevent the gelation of the solution prior to 3D bioprinting. Once the liquid constructs are 3D bioprinted, gelation of the constructs is induced when incubated at 37 °C for one hour [81,155,156].

The advantages of using the 3D bioprinting seeding technique include the automated approach for fabrication of the reproducible, complex 3D tissue microstructures that could biomimic the native physiological environment in various sizes and shapes. Furthermore, the deposition of multiple cell types, biomaterials, and biomolecules in precise locations in a layer-by-layer assembly with homogenous distribution of cells across the tissue constructs in a timely manner distinguishes this technology from the conventional fabrication methods [157]. However, the limitations of this technology include the inability to fabricate branched, perfused vascularised networks within the tissue engineered scaffold of various sizes and shapes to guarantee the long-term viability of the constructs. Furthermore, the inability to fabricate multifaceted patterning of heterocellular tissues and the challenge of maintaining cell viability and long-term functionality post-printing until remodelling and regeneration of the defected tissue/organ is a drawback (Table 2).

3D bioprinting has been explored for the fabrication of heart valves cellularised with tissue-specific cells, such as aortic root sinus (ARS) SMCs, and aortic valvular interstitial cells (AV-VICs) [158] or MSCs [159]. One in vitro study reported on the fabrication of 3D bioprinted aortic valve conduits composed of alginate/gelatin hybrid hydrogel encapsulated by ARS-SMCs and AV-VICs, finding biomimicry of the native aortic microenvironment in terms of similarity in anatomical structure and cellular composition [158]. The results showed cell’s viability (>80%) over a 7-day culture period, increased expression of alpha-smooth muscle actin by SMCs, and increased expression of vimentin by VICs. Another study by the same group discussed 3D bioprinted heart trileaflet valve conduits composed of methacrylated hyaluronic acid (Me-HA) and gelatin-methacrylate (GelMa) hydrogels encapsulated by AV-VICs [160]. Results showed optimisation of each hydrogel was crucial to print fidelity, thereby increasing the viability of encapsulated cells and remodelling of the original matrix by depositing collagen and GAGs.

MSCs encapsulated in GelMa/poly (eythelene glycol) diacrylate hydrogels supported by polycaprolactone were bioprinted to fabricate an anatomically accurate paediatric aortic heart valve with the ability to grow and remodel [159]. Assessments based on the production of collagen matrix and valve haemodynamics under physiological conditions showed there was an increase in collagen type I production under pulsatile shear stress conditions, and haemodynamic readings comparable with commercial valves.

An in vitro validation was also performed on 3D bioprinted cardiac tissue grafts from human induced pluripotent stem cells (hiPSCs)-derived cardiomyocytes cell spheroids [161]. Biomechanical conditioning based on the static mechanical stretching of the grafts showed significant elevation in sarcomeric length in comparison to unstimulated free-floating tissues, reduced elastic modulus, enhanced maximal contractile force, and elevated alignment in the formation of the ECM. Furthermore, stretched tissues exhibited an upregulation of genes expressed by cardiac-specific gene transcripts confirming cardiac-like cellular identity and enhanced remodelling by surrounding cardiac fibroblasts.

Bejleri et al. reported the use of a 3D bioprinted patch consisting of a hybrid bioink (cardiac dECM hydrogel and GelMA) loaded with paediatric human cardiac progenitor cells (hCPCs) in the potential repair of damaged myocardium associated with CHD [155]. In vitro results showed that hCPCs maintained above 75% viability in addition to a 30-fold increase in cardiogenic gene expression of hCPCs due to the presence of cardiac ECM in the matrix as opposed to stand-alone GelMA patch. Furthermore, elevated angiogenic potential (>2-fold) and enhanced endothelial cell tube formation were exhibited by the cardiac dECM-GelMA patch. In vivo results suggested that the patch was preserved with evidence of vascularisation during the 14 days period in rats’ hearts. Another animal study carried out functionality and histological assessments of two engineered heart tissue structures (EHTS) in the treatment of CHD [162]. The first structure was cardiac organoids (COs), composed of hiPSC-derived cardiomyocytes, human umbilical vein endothelial cells, and human fibroblasts. The second bioprinted tubular EHTS composed of COs were beating. A comparison between both structures was made based on their implantation around the aorta and inferior vena cava and subcutaneous injection on the back of the mice. Tubular EHTS displayed superior results compared with the COs, beating 1-month post-surgery and possessing myocardium striation and vascularisation within the tubular EHT matrix.

The application of cardiac bioinks with optimal chemical-mechanical characteristics that best recapitulate the cardiac microenvironment has not yet been fulfilled [163]. The development of synchronous contractile function between the engineered cardiac tissue and host remains a challenge due to the suboptimal electrical properties of cells and tissues such as cardiac excitation-contraction coupling, calcium transients, and cell–cell interactions [164]. There are few reports on the use of 3D bioprinting techniques to fabricate high-definition, anatomically accurate scaffolds for cardiac reconstructive surgery in paediatric CHD patients. As such, further research must be conducted, especially since this is a promising field for the future of personalised medicine through the use of patient-specific data and biological materials.

**Table 3 bioengineering-10-00057-t003:** Summary of cell types used for cellularisation of biological scaffolds.

Cell Type	Application	Seeding Method	Biological Scaffold
Peripheral mononuclear cells	Human use: paediatric patient	Manual	Decellularised pulmonary heart valve allograft [140]
hiPSC-cardiomyocytes and cardiac fibroblast	In vitro	Manual	dECM [141]
hiPSCs-cardiomyocytes	In vitro	Bioprinting	3D bioprinted cardiac tissue grafts from cell spheroids [161]
hiPSC-cardiomyocytes, HUVECs, and fibroblast	In vivo	Bioprinting	Cardiac Organoids forming EHTS [162]
hUCB-MSCs	In vivo	Manual	AM-based scaffold [133]
MSCs	In vivo	Manual	SIS [148]
MSCs	In vivo	Manual	Ultra-foam collagen sponge [149]
MSCs	In vitro	Bioprinting	GelMa/poly (eythelene glycol) diacrylate hydrogels supported by PCL [159]
U-SCs	In vivo	Manual	Polyurethane/SIS patch [144]
RAECs and RMCs	In vitro	Manual	Hybrid HPM [143]
PCs	In vitro	Manual	Decellularised porcine pulmonary artery [145]
PCs	In vitro	Manual	Decellularised cardiac ECM [146]
EPCs	In vitro	Manual	Decellularised aortic heart valve [147]
hCPCs	In vivo	Bioprinting	Patch consisting of a hybrid bioink made of cardiac dECM hydrogel and GelMA loaded with paediatric hCPCs [155]
Pericytes	In vivo	Manual	Proxicor^®^ [2]
SMCs	In vitro	Manual	Reinforced porcine blood-derived fibrinogen [142]
SMCs	In vivo	Manual	Decellularised swine SIS graft [80]
SMCs	In vivo	Manual	Scaffold-free. Ten-layered cell sheets of SMCs [150]
ARS-SMCs and AV-VICs	In vitro	Bioprinting	Aortic valve conduits composed of alginate/gelatin hybrid hydrogel [158]
AV-VICs	In vitro	Bioprinting	Heart trileaflet valve conduits are composed of Me-HA and GelMa hydrogels [160]
Neonatal cardiac Fibroblast	In vitro	Injection	Direct injection encapsulating with a 3D fibrin gel artificial heart muscle patch [151]
Neonatal cardiac Fibroblast	In vitro	Injection	Direct injecting into the novel BECV [152]

Abbreviations: hiPSCs-human induced pluripotent stem cells; HUVECs-human umbilical vein endothelial cells; hUCB-MSCs-human umbilical cord blood-derived mesenchymal stem cells; MSCs-mesenchymal stem cells; U-SCs–urine-derived stem cells; RAEC-rat aortic endothelial cells; RMC-rat bone marrow cells; PC-progenitor cell; EPC-endothelial progenitor cells; hCPC-human cardiac progenitor cells; SMC-smooth muscle cells; ARS-SMC-aortic root sinus smooth muscle cells; AV-VIC-aortic valvular interstitial cells; Me-HA-methacrylated hyaluronic acid; GelMa-gelatin-methacrylate; EHTS-engineered heart tissues structures; BECV-bioengineered complete ventricle; PCL-polycaprolactone; HPM- homogenised pericardium matrix.

## 5. Future Work and Translation to Clinic

The clinical translation of fabricated tissue-engineered scaffolds exhibiting growth potential and remodelling characteristics for reconstructive surgeries for paediatric CHD patients has been slow over the last few years [165,166]. The main challenge facing this field is the fabrication of tissue-engineered scaffolds (including whole hearts) with high precision, intricacy, stability, and minimised host immune rejection, as well as consideration of regulatory concerns when implanting in growing paediatric patients with CHD [167,168]. It is difficult to concomitantly possess all these characteristics, especially since scaffolds must exhibit a stable structure with high integrity and functionality when exposed to the native physiological environment upon implantation [167]. This could ultimately, over time, affect the balance between de novo ECM production and scaffold degradation. Furthermore, complete decellularisation is difficult to achieve and foreign tissue can cause residual immunogenicity leading to degeneration and stenosis of the scaffold, potentially leading to surgical reintervention to replace the implant [168]. With regards to regulatory concerns, quality control of the tissue-engineered products remains a challenge, in addition to product consistency between batches and reliable preclinical testing. There is room for potential growth in the field of genetic engineering when coupled with cardiac regenerative medicine to produce donor hearts from alternative sources [165]. A recent report of the first successful pig-to-human clinical xenotransplantation made huge headlines [169]. However, the patient who previously had end-stage heart disease survived for only two months post-surgery due to the immune rejection response [170]. The donor pig underwent ten genetic manipulations: knockdown of three immune rejection-related porcine genes, and insertion of six human genes and one growth gene necessary for manipulating the heart size. Further studies must be implemented to understand how effective the current immunosuppressants are in controlling the host rejection response to xenografts in addition to the spreading of pathogens. Ethical considerations around pig-to-human xenotransplantation are also complex.

When considering specific applications for paediatric CHD patients, the growth potential of genetically engineered scaffolds with somatic growth is a key consideration. Considering acellular or recellularised biological scaffolds, there are advantages and disadvantages associated with each (Table 1). Acellular tissue-engineered scaffolds need to be suturable, biodegradable scaffolds coupled with biological factors necessary for angiogenic, immunomodulatory, and anti-inflammatory responses. Due to the low proliferation rate of cardiomyocytes, recellularised tissue-engineered scaffolds might be better suited for paediatric application. Such constructs could have low density of proliferative (primary or stem cell-derived) cardiomyocytes, cardiac fibroblasts, and endothelial cells that are able to survive and function due to the good accessibility to nutrients, oxygen, and/or blood before the development of good vasculature. Alternatively, recellularised scaffolds could possess an elevated, physiologically relevant density of the above-mentioned cell types present within integrated perfusable vascularised channels undergoing quick anastomosis in vivo [171]. Built-in vascularised structures could be further matured into biomimic vessel-like structures in a bioreactor [172]. To avoid the invasive approach required to extract autologous cells from patients, stem cells from non-invasive sources can be used. However, the optimisation of the differentiation protocols of stem cells to cardiomyocytes, endothelial cells, and SMCs remains a challenge.

In terms of scaffold biodegradability, degradation waste products should be body-friendly in addition to being used as an immunomodulatory tool that can trigger enhanced healing and integration with the host tissue [171]. This is done through the modulation of cell recruitment, infiltration, and activation of inflammatory pathways, thereby causing positive control of the inflammatory response. Furthermore, the rate of integration of the tissue-engineered scaffold with the host tissue must outperform the degradation rate of the scaffold to avoid scaffold collapse during remodelling.

A better understanding of the process from implantation of the tissue-engineered scaffold to functional new tissue formation in the host could be achieved by advanced computational modelling and detailed animal studies. Through the inclusion of various input parameters, such as connections between different cell types, implanted scaffold factors, and cytokines, in computational modelling software, a better prediction of the combined effect of these inputs and the host environment on new tissue formation could be obtained. In an effort to improve clinical outcome, prediction of the effect of changes in scaffold parameters could be explored computationally in the short and long term [39]. When computational modelling is coupled with 3D printing and 3D imaging, this will lead the way towards personalised medicine using patient-specific data and information to treat congenital malformations [173]. Observations on the application of 3D bioprinting to develop cell-laden scaffolds for use in paediatric CHD treatment are very limited, and this is an area of research with significant promise.

There is little progress in terms of granting regulatory approval for clinically applicable tissue-engineered products, mainly due to a lack of consensus on classification and suitable surveillance programs. The classification used to define implanted materials and testing techniques falsely forecasts the efficacy of the tissue-engineered products. It is important for regulatory bodies to ease the process by standardising the pipeline and consequently aid the acquisition of approvals necessary to accelerate the safe, efficient use of tissue-engineered products in national and international markets [39,174].

## 6. Conclusions

It is imperative that the field of TE continues to grow and progress until palliative interventions evolve into curative treatments. In order for this to happen, technological advances and a deeper understanding of the mechanisms involved in the host environment are needed. This will allow the design and fabrication of scaffolds that recapitulate the native environment, with a particular focus on the safety and efficacy of those products in preclinical and clinical settings. Such scaffolds must possess growth potential and suitable haemodynamic characteristics, along with a resistance to thrombosis, calcification, inflammation, and stenosis. To overcome the several challenges associated with the clinical translation of tissue-engineered products from bench to bedside, experts from various fields, including clinicians, biomedical engineers, software engineers, and biologists, among others, must work collaboratively.

## Figures and Tables

**Figure 1 bioengineering-10-00057-f001:**
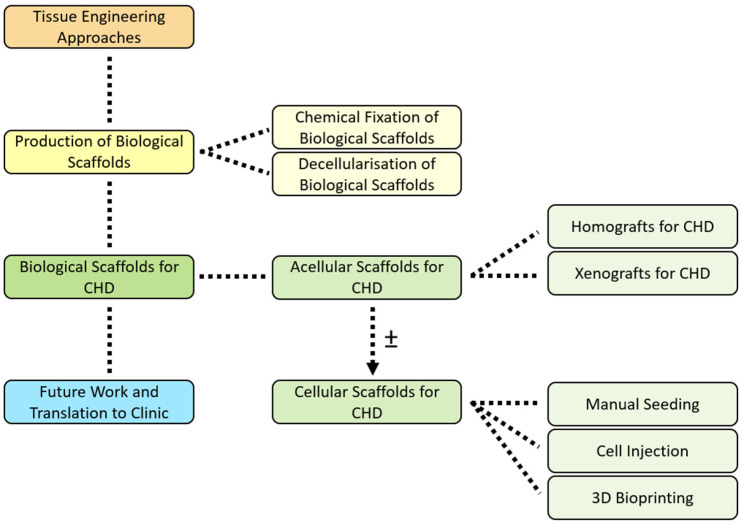
Schematic of review structure.

**Figure 2 bioengineering-10-00057-f002:**
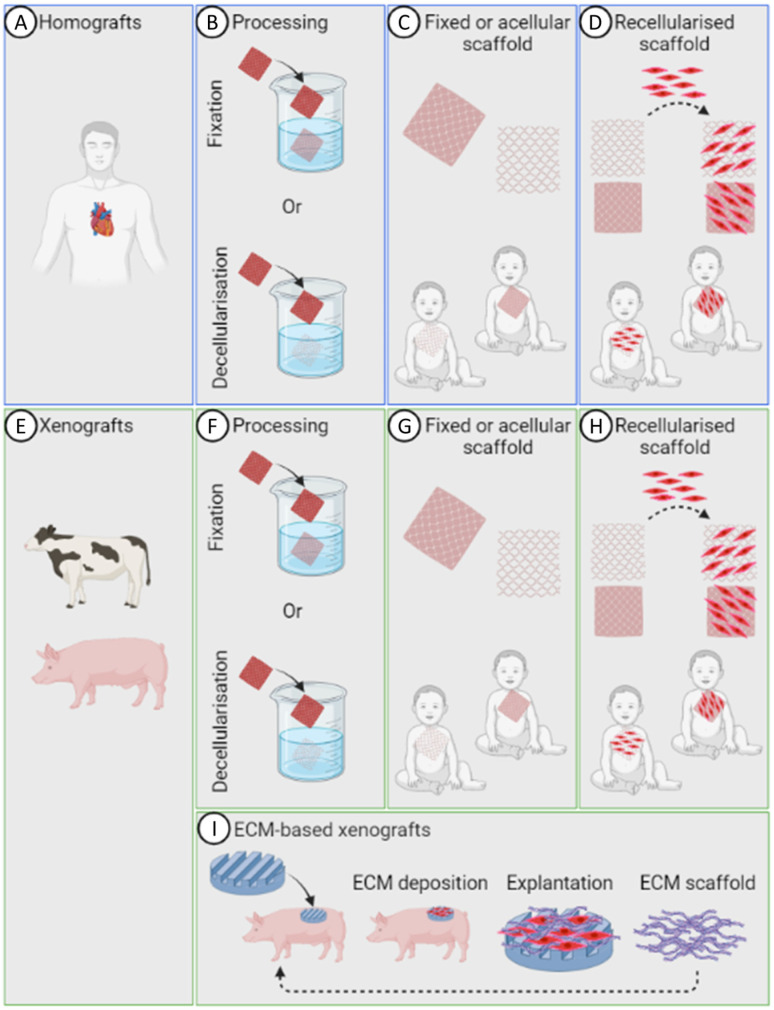
Schematic representation of biological scaffold production for CHD. (**A**) Tissues from human (patient donors or cadavers) origin are collected to be used as homografts. (**B**) Tissues require processing by either fixation or decellularisation to reduce antigenicity and avoid rejection by the host. (**C**) Processed materials can be used as fixed or acellular scaffolds in the clinic, or recellularised (**D**) with autologous or allogenic cells prior to implantation in CHD patients. (**E**) Tissues animal (e.g., bovine, swine, ovine, etc.) origin are collected to be used as xenografts. (**F**) Tissues require processing by either fixation or decellularisation to reduce antigenicity and avoid rejection by the host. (**G**) Processed materials can be used as fixed or acellular scaffolds in the clinic, or recellularised. (**H**) with autologous or allogenic cells prior to implantation in CHD patients. (**I**) Production of ECM-based grafts involves percutaneous implantation of a synthetic non-biodegradable scaffold, which acts as a mould for ECM deposition. The mould with the deposited tissue will be explanted and the ECM, having been physically separated from the mould, used as an autologous ECM-based graft in the original host.

**Figure 3 bioengineering-10-00057-f003:**
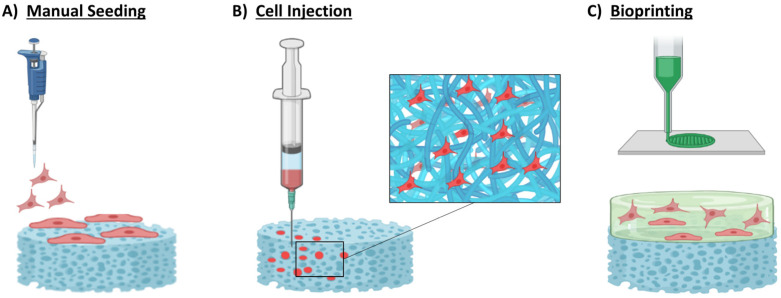
Schematic representation of techniques for biological scaffold cellularisation. (**A**) Manual seeding: cells resuspended in a fluid solution are deposited on the surface of the scaffold and cultured for maturation. (**B**) Cell Injection: Cells are injected throughout the thickness of the matrix creating a fully colonised graft. (**C**) Bioprinting: Cells are encapsulated in a biocompatible hydrogel and extruded with high accuracy on the surface of the graft providing specific spatial organisation.

**Table 1 bioengineering-10-00057-t001:** Summary of the discussed advantages and disadvantages of homograft, xenograft, acellular, and cellular scaffolds.

	Advantages	Disadvantages
Homografts	Capacity for growth and repair. Autografts are non-immunogenic. Suitable haemodynamics. Manipulability in surgery. Native tissue properties.	Limited availability, particularly in the context of paediatric CHD. Autologous pericardium degenerates due to mechanical stress of the arterial environment. Allografts are immunogenic, though less so than xenografts. Aldehyde fixation can lead to cytotoxicity.
Xenografts	Good availability of off-the-shelf commercial products ready for implantation.	Significant immunogenic concerns necessitating removal or masking of xenoantigens to avoid severe immune rejection. Aldehyde fixation can lead to cytotoxicity.
Acellular	Good availability of off-the-shelf commercial products ready for implantation.	Unable to grow immediately upon implant, first requiring endogenous cell recruitment and infiltration of the scaffold in the patient.
Recellularised	Growth and remodelling capacity. Personalised medicine approach if autologous cells are used, supporting immunocompatibility. Potential to recapitulate native properties of the tissue.	Technically challenging. Time-consuming, preventing immediate access to off-the-shelf products. Harvesting cells can be invasive, depending on the source.

**Table 2 bioengineering-10-00057-t002:** Summary of the discussed advantages and disadvantages of different cellularisation techniques: manual seeding, cell injection, and 3D bioprinting.

	Advantages	Disadvantages
Manual Seeding	Inexpensive. Simple and widely used technique.	Heterogenous distribution of cells on the scaffold. Difficulty in controlling the localisation of cells on the scaffold. Inefficient and impractical fabrication tool.
Cell Injection	Simple technique.	Inconsistency of cell suspension across the layered scaffold. Exposure of cells to high shear rate/pressure. Slow and time-consuming process. Might be suitable for small-scale seeding.
3D bioprinting	Homogenous distribution of cells on the scaffold. Precise deposition of multiple cell types, biomaterials, and biomolecules on a layer-by-layer basis to create 3D structures that resemble the native physiological environment. Ability to print various geometries (simple/complex) with high precision, intricacy, and enhanced resolution.	Difficulty in developing vascularised networks at a single cell level. Difficulty in achieving multifaceted patterning of heterocellular tissues. Inability to preserve cell viability and long-term functionality post-printing until remodelling and regeneration of the defected tissue/organ are achieved.

## Data Availability

Not applicable.

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
