# Peer review of "Biological Scaffolds for Congenital Heart Disease"

_bioengineering, 2023, doi:10.3390/bioengineering10010057_

Round 1

Reviewer 1 Report

This review mainly discussed the existing tissue engineering approaches and biological scaffold fabrication techniques, as well as the application of biological scaffolds and the existing problems and challenges of biological scaffolds for congenital heart disease. It needs to be further strengthened in grasping the theme, the overall structure, and the logic between sentences.

1. Please specify what "many interventions" in Paragraph 3 of the Introduction are because many new techniques mentioned in Paragraph 4 are also clinically counted as interventional therapy.

2. The authors mentioned "imperfect" in the fourth paragraph of the Introduction. Please specify what it is.

3. "the search for readily available and biocompatible replacement parts endowed with growth and adaptive remodeling capacity, as well as durability over the patient's lifetime", why do patients with congenital heart disease need such biomaterials?

4. It is suggested that the authors could echo the theme "Biological Scaffolds for Congenital Heart Disease" of this article in every part. The authors only write about biological scaffolds for CHD in the fourth part and the application prospect in the fifth part, but they discussed other contents in the second and third parts, which makes the structure of this article seem confusing.

5. In 4.1, the authors discussed the biological bioactive materials and briefly listed the uses of acellular scaffolds, but there is little reference to CHD. The description deviated from the theme of biological scaffolds for CHD in part IV.

6. In 4.2, the authors mainly discussed the preparation techniques of cellular scaffolds, such as 4.2.1, 4.2.2, and 4.2.3, but rarely involve biological scaffolds for CHD.

7. "The main challenge facing this field is the fabrication of tissue engineering scaffolds (including whole hearts) with high precision, intricacy, durability, and minimized host immune rejection." Please add references and elaborate on this paragraph, as this is also what the reader is concerned about.

8. What is the relationship between "In terms of the biodegradability of the scaffolds……" and "a better understanding of the process from implantation of the tissue engineering scaffold……"It is suggested that there is only one center in a paragraph, and the center is explained in detail.

Author Response

Reviewer # 1

This review mainly discussed the existing tissue engineering approaches and biological scaffold fabrication techniques, as well as the application of biological scaffolds and the existing problems and challenges of biological scaffolds for congenital heart disease. It needs to be further strengthened in grasping the theme, the overall structure, and the logic between sentences:

  1. Please specify what "many interventions" in Paragraph 3 of the Introduction are because many new techniques mentioned in Paragraph 4 are also clinically counted as interventional therapy.

RESPONSE: We extended the paragraph 3 of the introduction presenting examples of palliative procedures. In addition, we clarified what a palliative intervention for paediatric patient with congenital heart disease (CHD) is, with the aim to avoid further misunderstanding.

In fact, the same procedure could be used for both palliative and curative interventions (interventional therapy procedure), but the final goal distinguish them. Curative procedure aims to repair the diseased heart restoring physiological condition, while the palliative one correct the disease for a limited period as mentioned in the manuscript “controlling the heart failure and preparing for a later correction when the pediatric patient grows to a suitable and stable condition”.

  1. The authors mentioned "imperfect" in the fourth paragraph of the Introduction. Please specify what it is.

RESPONSE: The wording of previous paragraph 4 has been modified to address the critic and extended (now from line 50 to line 56 of the introduction section) specifying the limit and imperfection of the traditional methodologies.

  1. "the search for readily available and biocompatible replacement parts endowed with growth and adaptive remodeling capacity, as well as durability over the patient's lifetime", why do patients with congenital heart disease need such biomaterials?

RESPONSE: To highlight the importance of having grafts able to grow together with the paediatric CHD patient, we provided additional detail line 57 to 64. The anatomy of the diseased heart will modify during the natural somatic growth of the young patient, leading to failure of implanted prosthetic graft and multiple surgery that drastically reduce the quality of life.

  1. It is suggested that the authors could echo the theme "Biological Scaffolds for Congenital Heart Disease" of this article in every part. The authors only write about biological scaffolds for CHD in the fourth part and the application prospect in the fifth part, but they discussed other contents in the second and third parts, which makes the structure of this article seem confusing.

RESPONSE: We are thankful to the reviewer for this suggestion. We amended removing the section that was describing the synthetic materials to avoid misleading the reader from the main topic of the review.

  1. In 4.1, the authors discussed the biological bioactive materials and briefly listed the uses of acellular scaffolds, but there is little reference to CHD. The description deviated from the theme of biological scaffolds for CHD in part IV.

RESPONSE: We apologize if the terminology present in the text created doubt in the reader. The whole review focuses on the detailed analysis of the existing biological scaffold used in paediatric CHD patients. Both acellular and cellular scaffolds are in fact part of the big category of biological scaffolds, as shown in Figure 1. We amended the text making sure that it will be clear for the reader that we always refer to biological scaffold for CHD.

  1. In 4.2, the authors mainly discussed the preparation techniques of cellular scaffolds, such as 4.2.1, 4.2.2, and 4.2.3, but rarely involve biological scaffolds for CHD.

RESPONSE: We addressed the comment of the reviewer following the same approach of the previous comment, clarifying that the meaning of the review is focusing on biological scaffolds. In fact, the cellularisation methodologies listed in this review have always as subject a biological scaffold.

  1. "The main challenge facing this field is the fabrication of tissue engineering scaffolds (including whole hearts) with high precision, intricacy, durability, and minimized host immune rejection." Please add references and elaborate on this paragraph, as this is also what the reader is concerned about.

RESPONSE: We extended the section 5 of the review and added references supporting the discussion.

  1. What is the relationship between "In terms of the biodegradability of the scaffolds……" and "a better understanding of the process from implantation of the tissue engineering scaffold……",It is suggested that there is only one center in a paragraph, and the center is explained in detail.

RESPONSE: We addressed the comment of the reviewer providing a wider explanation of the underlined topic.

Reviewer 2 Report

The manuscript titled “Biological Scaffolds for Congenital Heart Disease” by Harris et al. is a well-written review paper mainly summarized the current technical challenges and potential solutions of tissue engineering strategies in the treatment of pediatric CHD. The objective of this review is of significance in the field of tissue engineering. By presenting the flaws of the current clinical intervention in treating CHD, the authors discussed the advantages of tissue engineering approaches and particularly those involving biological scaffolds. Following the introduction of scaffold fabrication techniques, the authors then summarized both the acellular and cellular scaffolds for CHD treatment, especially the commercially available ones. The overall flow is good, and I enjoyed reading through the manuscript.

A minor revision is recommended. Before it can be accepted for publication, I have some small suggestions that are noted below.

1.      Please add some tables to compare the pros and cons of different scaffolds and different cellularization methods to help readers understand easier.

2.      Please summarize more recent studies. I understand that the publications about commercialized products cannot be the most updated ones. However, the overall percentage of references from 2021/2022 is only ~10%, which is still low as a review paper.

3.      There are some mixed uses of “biological scaffolds” and “tissue engineering scaffolds”. Biological scaffolds are those made of ECM-based materials while TE scaffolds also include those made of other materials such as metal/ceramics/synthetic polymers. Since this review mainly discussed the fabrication techniques and classification of ECM-based scaffolds, I would suggest the authors focus on biological scaffolds only.

Author Response

Reviewer # 2

The manuscript titled “Biological Scaffolds for Congenital Heart Disease” by Harris et al. is a well-written review paper mainly summarized the current technical challenges and potential solutions of tissue engineering strategies in the treatment of pediatric CHD. The objective of this review is of significance in the field of tissue engineering. By presenting the flaws of the current clinical intervention in treating CHD, the authors discussed the advantages of tissue engineering approaches and particularly those involving biological scaffolds. Following the introduction of scaffold fabrication techniques, the authors then summarized both the acellular and cellular scaffolds for CHD treatment, especially the commercially available ones. The overall flow is good, and I enjoyed reading through the manuscript. A minor revision is recommended. Before it can be accepted for publication, I have some small suggestions that are noted below:

  1. Please add some tables to compare the pros and cons of different scaffolds and different cellularization methods to help readers understand easier.

RESPONSE: We are thankful to the reviewer for this suggestion. We have introduced in the text two new tables respectively focused on: Table 1)  Summary of the discussed advantages and disadvantages of homograft, xenograft, acellular, and cellular scaffolds; Table 3) Summary of the discussed advantages and disadvantages of the various cellularization techniques: manual seeding, cell injection, and 3D printing.

  1. Please summarize more recent studies. I understand that the publications about commercialized products cannot be the most updated ones. However, the overall percentage of references from 2021/2022 is only ~10%, which is still low as a review paper?

RESPONSE: We performed more analytic research of the literature looking for more recent publication in the area of biological scaffolds used for CHD, improving our percentage to 20% 2021/2022 and above 30% 2020/2022.  

  1. There are some mixed uses of “biological scaffolds” and “tissue engineering scaffolds”. Biological scaffolds are those made of ECM-based materials while TE scaffolds also include those made of other materials such as metal/ceramics/synthetic polymers. Since this review mainly discussed the fabrication techniques and classification of ECM-based scaffolds, I would suggest the authors focus on biological scaffolds only.

RESPONSE: We amended removing the section that was describing the synthetic materials to avoid misleading the reader from the main topic of the review.

Reviewer 3 Report

This review summarized the scaffolds for congenital heart diseases treatment. Particularly the authors focused on biological scaffolds, mainly decellularized ECM. They described clinical results using biological scaffolds. Overall, the topic will be interesting for the readers in medical and material researchers and the contents are well-summarized. However, it is difficult to understand the paper due to providing the figures and tables for the explanation insufficiently. Therefore, I recommend the authors to revise the manuscript for the publication. The specific comments are below.

1)     In Figure 2, it looks that the decellularization is performed after fixation. If the authors want to show that fixed and decellularized tissues are used in clinic, the figure should be revised to avoid such misunderstanding. Also, the figure title in the legend includes “fabrication technique”. But the figure does not contain fabrication technique (or contain a part of fabrication technique). The authors should revise the figure title to clearer one.

2)     In section 4, the authors described the examples to use decellularized ECM for CHD treatments. But I cannot understand the differences between acellular scaffolds and ECM-based graft and the reasons why the authors divided subsection 4.1 into “Homografts”, “Xenografts”, and “ECM-based grafts” (if the ECM is derived from animals, is it xenografts?). I think application types are clearly classified. I felt that the figure would be helpful for the understanding.

3)     Please add the table to summarize previously-reported dECM for CHD treatment with dECM’s origins of species. Maybe, the table will be inserted in the start of section 4.

4)     Please summarize cell types for CHD treatments with dECM in a table in section 4.2.

5)     The title of section 4.2.2. should be “Injection Technique”.

6)     In section 4.2.3., the authors summarized the dECM for 3D bioprinting. But such kinds of dECM should be solubilized by some methods, which is different from previous sections. And the explanation of solubilization is lacked in the text. So, the authors should be described the necessity of solubilization and the methods briefly.

Author Response

Reviewer # 3

This review summarized the scaffolds for congenital heart diseases treatment. Particularly the authors focused on biological scaffolds, mainly decellularized ECM. They described clinical results using biological scaffolds. Overall, the topic will be interesting for the readers in medical and material researchers and the contents are well-summarized. However, it is difficult to understand the paper due to providing the figures and tables for the explanation insufficiently. Therefore, I recommend the authors to revise the manuscript for the publication. The specific comments are below.

  1. In Figure 2, it looks that the decellularization is performed after fixation. If the authors want to show that fixed and decellularized tissues are used in clinic, the figure should be revised to avoid such misunderstanding. Also, the figure title in the legend includes “fabrication technique”. But the figure does not contain fabrication technique (or contain a part of fabrication technique). The authors should revise the figure title to clearer one.

RESPONSE: We are thankful to the reviewer for pointing out the unclarity of the message given by the figure 2. We have amended the Figure 2 to reflect exactly what is reported in the text. In addition, we modified the title of the figure to “the production of biological scaffolds for CHD”.

  1. In section 4, the authors described the examples to use decellularized ECM for CHD treatments. But I cannot understand the differences between acellular scaffolds and ECM-based graft and the reasons why the authors divided subsection 4.1 into “Homografts”, “Xenografts”, and “ECM-based grafts” (if the ECM is derived from animals, is it xenografts?). I think application types are clearly classified. I felt that the figure would be helpful for the understanding.

RESPONSE: We have modified the text following the reviewer suggestion. Specifically, we included the “ECM-based grafts” derived by percutaneous implantation in the Xenograft section and we modified accordingly the figure 2 (as mentioned in the previous point).

  1. Please add the table to summarize previously-reported dECM for CHD treatment with dECM’s origins of species. Maybe, the table will be inserted in the start of section 4.

RESPONSE: We appreciate the reviewer comment. However, we believe that creating this type of table will mean including the majority of the study reported, leaving few of them out. We would like to keep equal importance on all type of biological scaffolds.

  1. Please summarize cell types for CHD treatments with dECM in a table in section 4.2.

RESPONSE: We have created the table (Table 2) reporting all the cell types used for the cellularization techniques.

  1. The title of section 4.2.2. should be “Injection Technique”.

RESPONSE: We have changed the title, but in our opinion “Cell injection” would be more appropriate.

  1. In section 4.2.3., the authors summarized the dECM for 3D bioprinting. But such kinds of dECM should be solubilized by some methods, which is different from previous sections. And the explanation of solubilization is lacked in the text. So, the authors should be described the necessity of solubilization and the methods briefly

RESPONSE: Following the reviewer comment, we expanded the section 4.2.3 explaining the solubilization methodologies and rationale.

Round 2

Reviewer 1 Report

The manuscript is ready for publication.

Reviewer 3 Report

The authors revised properly. So, I recommend to publish this manuscript as it is.